

Parallel functional and stoichiometric trait shifts in South-American and African forest
communities with elevation
Marijn BAUTERS[1, 2,†], Hans VERBEECK[2], Miro DEMOL[2], Stijn BRUNEEL[2], Cys
TAVEIRNE[1], Dries VAN DER HEYDEN[1], Landry CIZUNGU[3] and Pascal BOECKX [1]
[1] Isotope Bioscience Laboratory – ISOFYS, Ghent University, Coupure Links 653, 9000 Gent,
Belgium
[2] CAVElab, Computational and Applied Vegetation Ecology, Department of Applied Ecology
and Environmental Biology, Ghent University, Coupure Links 653, 9000 Ghent, Belgium
[3] Faculty of Agronomy, Université Catholique de Bukavu, Avenue de la mission, BP 285,
Bukavu, DR Congo
[†] Corresponding author, e-mail: Marijn.Bauters@UGent.be
Co-authors : Hans.Verbeeck@UGent.be, mirodemol@hotmail.com, Stijn.Bruneel@UGent.be,
cysenco@hotmail.com, driesvdheyden@gmail.com, landrycizunngu@yahoo.fr,
Pascal.Boeckx@UGent.be



Abstract
Elevational gradients are an empirical tool to assess long-term forest responses to environmental
change. We studied whether functional composition of tropical forest along elevational gradients
in South America and in Africa showed similar shifts. We assessed community-weighted
functional canopy traits and indicative $\delta^{15}N$ shifts along two new altitudinal transects in the tropical
forest biome of both South-America and Africa. We found that the functional forest composition
response along both transects was parallel, with a species shift towards more nitrogen conservative
species at higher elevations. Moreover, canopy and topsoil $\delta^{15}N$ signals decreased with increasing
altitude, suggesting a more conservative N cycle at higher elevations. This cross-continental study
provides two empirical indications that both South-American and African tropical forest show a
parallel response along altitude, driven by nitrogen availability along the altitudinal gradients,
inducing a parallel shift in the functional forest composition. This highlights the importance of
nutrient availability for tropical forest in a changing world. More standardized research, and more
research on other elevational gradients is needed to confirm our observations.





## Introduction

A good understanding of the future response of tropical forest ecosystems to global changes is required because of their vital role in the global carbon cycle. Recent efforts have shown that biosphere-atmosphere carbon exchange in forests is regulated by nutrient availability (Fernandez-Martinez et al., 2014). Therefore, changes in nutrient bio-availability induced by global change need to be taken into account. Uncertainties on future responses of forest ecosystems to global change are perhaps most prominent in the tropics, where monitoring is underdeveloped compared to the temperate region. Recent work has shown that African and South-American tropical forest show important differences in structure (Banin et al., 2012) and species richness and composition (Slik et al., 2015a). Additionally, there is a lack of cross-continental empirical research in both the Amazon and the Congo basin, and most model projections are parameterized on either South-American tropical or even temperate forests (Corlett and Primack, 2006). Moreover, earth system model projections typically do not account for nutrient dynamics (Anav et al., 2013) and models that account for nutrient cycling in their land component are still under development (Goll et al., 2012; Reed et al., 2015; Smith et al., 2014). , In this context we can raise questions about the universality of tropical forest biogeochemistry and functioning across both continents, and subsequently their response to future global change scenarios. This leaves us with an important bottleneck to further optimize Earth system models (Ciais et al., 2011). Hence, cross-continental comparisons of tropical forest ecosystem responses to environmental gradients are needed. Additionally, due to the central role of nutrient availability that drives both net ecosystem productivity (NEP) and ecosystem carbon use efficiency ($CUE_e$) (Fernandez-Martinez et al., 2014), the effect of climatic gradients on nutrient availability should be better understood. Elevation transects offer such gradients.






Elevation transects have been postulated as a viable and useful setup to assess long-term ecosystem
responses to environmental changes, and hence serve as an empirical tool to assess future
trajectories of forest ecosystems under global change (Malhi et al., 2010; Sundqvist et al., 2013).
In the tropics, this has invoked research efforts on transects in South America, but no such studies
have been carried out in central African forests.

In this study we address this lack of standardized cross-continental research and assessed shifts in
nutrient availability and forest functional composition along two similar transects in Ecuador and
Rwanda. We assessed these shifts through indicative (I) community-level functional traits and (II)
nitrogen isotope ratios in topsoil and canopy. Canopy chemistry has received increasingly more
attention, and has been identified as a proxy for landscape-scale biogeochemistry (Asner et al.,
2015; Fyllas et al., 2009), because of its inherent link to the plant strategy. Nevertheless, and as
rightfully noted by Asner and Martin (2016), there are only limited surveys on canopy functional
signatures in the tropics, while this information is vital for a landscape-scale understanding of
tropical forest assembly. In addition to a standard set of leaf traits, both leaf and soil $\delta^{15}N$ are
known integrators of the local N-cycle. Previous efforts have shown that shifts towards lower $\delta^{15}N$
values indicate a more closed N-cycle with lower N availability, and *vice versa* (Craine et al.,
2009, 2015). Hence combining both leaf traits and $\delta^{15}N$ values is an interesting approach to assess
ecosystem responses to environmental gradients. We hypothesized that (I) both these community-
level traits and stable isotope signals would indicate a shift in nitrogen availability with altitude,
and that (II) these shifts would be similar in terms of direction and magnitude, given a standardized
research protocol and a similar adiabatic lapse rate.





Materials and Methods

**Field inventories, sampling and trait analyses**

We selected plots at different altitudes on the West flank of the Andes in Ecuador (ranging from 400-3200 masl) and in the Nyungwe national Park Rwanda (1600-3000 masl), in the Southern Great Rift Valley (figure S1 and Table S1 for location and overview maps). Due to reduced accessibility, the gradient in Rwanda was shorter than the South-American transect. We delineated and inventoried plots following an international standardized protocol for tropical forest inventories (RAINFOR, Malhi et al. 2002), with an adapted plot size of 40 by 40 m. In each plot, the diameter of all live stems with a diameter larger than 10 cm was measured at 1.3 m height and the trees were identified to species or genus level. Besides diameters also tree heights were measured, in order to estimate the aboveground carbon storage (AGC), using pan-tropical allometric relationships (Chave et al., 2014). The canopy of every plot was characterized by selecting the most abundant tree species, aiming at a sampling percentage of 80% of the basal area of the plots. For the selected species of all plots, we sampled mature leaves of a minimum of three individuals per species per plot using tree climbers. For most of the individuals we sampled fully sunlit leaves, but this was not always possible for the safety of the climbers, in which case we sampled partly shaded leaves under the top canopy. Previous work on altitudinal transects has shown that the vertical profile of leaves within a canopy has little effect on the trait values (Fisher et al., 2013). This sampling was carried out at the level of the total inventory. Additionally, composite samples of the topsoil (0-5 cm) were collected at five different places within each plot, and mixed per plot prior to drying. Soil and leaf samples were dried for 48 hours at 60°C. Roots were picked out of the soil samples before grinding and subsequently carbon (C), nitrogen (N) content and $\delta^{15}N$ of plant and soil samples were analyzed using an elemental analyzer (Automated



Nitrogen Carbon Analyser; ANCA-SL, SerCon, UK), interfaced with an Isotope Ratios Mass
Spectrometer (IRMS; 20-20, SerCon, UK). Leaf samples were dry-ashed at 550°C for 5.5 hours;
the ash was dissolved in 2M HCl solution and subsequently filtered through a P-free filter. The
aliquots where then analyzed for total P by AAS method No.G-103-93 Rev.2 (Multitest
MT7/MT8; Ryan and others 2001). SLAs were calculated by dividing the leaf areas of all the
sampled leafs per individual by their summed dry mass. Leaf areas were determined by either
photographing leafs with on white paper with a reference scale or by drawing leaf contours and
scanning the drawings. Both the scans and the pictures were processed using the ImageJ software
(Schneider et al., 2012). For one abundant species of the higher altitudes on the Rwandan transect
(*Podocarpus latifolius* (Thunb.) R.Br. ex Mirb.) we could not obtain good area estimates, so we
adopted SLA figures from literature (Midgley et al., 1995).

**Statistical analysis**
Average leaf trait values as specific leaf area (SLA), leaf nitrogen content (LNC), leaf
phosphorus content (LPC), $\delta^{15}$N, C:N and N:P ratio were calculated for every selected species,
based on the sample values for the different individuals of the species. Subsequently, to calculate
community-level traits and leaf $\delta^{15}$N per plot, we calculated a basal area weighted-average canopy
value and standard deviation using the species composition and the species averages, following
Asner et al. (2016b). Hence:

$$\bar{x}_w = \frac{\sum_{i=1}^{N} w_i \cdot x_i}{\sum_{i=1}^{N} x_i}$$

with $x_w$ the weighted value for trait x, $x_i$ the mean trait value for species i and $w_i$ the basal-area
based weight of that species the specific plot. Subsequently for the weighted standard deviations
($\sigma_w$):





$$\sigma_w = \sqrt{\dfrac{\sum_{i=1}^{N} w_i \cdot (x_i - \bar{x}_w)^2}{\dfrac{(N-1)\sum_{i=1}^{N} w_i}{N}}}$$


with N the number of nonzero weights.
The structure of the trait datasets was assessed qualitatively using Pearson correlation statistics.
Finally, we studied the relations between the different leaf traits and elevation using mixed effects
models for the different traits, with a random error structure. The plots were spatially clustered
around four altitudes on both transects, hence we introduced these altitudinal clusters as a random
effect, and treated altitude and transect as fixed effects. Models were then fitted using maximum
likelihood methods in the 'nlme' package in R (Pinheiro et al., 2013).  An interaction term for
transect (Rwanda or Ecuador) and elevation was tested using likelihood ratio test and was retained
in the final models when significant at the P<0.05 level. For reasons of linearity we used the inverse
C:N (hence rather N:C) in these analyses. For the statistical analysis, the R-software was used (R
Core Team, 2014).

Results
The pooled trait datasets from both transects showed a consistent and similar correlation structure
(Fig S2), with in both separate and the pooled data significant correlations between all traits, except
SLA and N:P. The structural vegetation parameters on both transects showed important
differences: for the same altitude range, we found a higher stem density, but less species on the
Rwandan transect (Table 1). Tree height and basal area were comparable, and the carbon stocks
showed high variability along both transects. Climatic conditions were similar, with a highly



consistent temperature gradient (Fig S3, Table 1), and similar mean annual precipitation in the
concurring altitudinal ranges. The linear mixed models with altitude as fixed effect, were able to
explain a significant proportion of variation in all traits. This is reflected by both the marginal and
conditional $R^2_{ajd}$, respectively proxies for the variation explained by the fixed effects, and the
random and fixed effects together (Schielzeth and Nakagawa, 2013) (Table 2). The interaction
term was not significant in any case, hence the trait and $\delta^{15}N$ responses to altitude were parallel on
both continents. LNC, N:C, LPC and N:P significantly decreased with altitude ($R^2_{adj,marg}$ of
respectively 0.83,0.87,0.68 and 0.60), with the Rwanda transect showing higher overall values.
SLA also decreased significantly, but with a slightly higher intercept for the Ecuadorian transect
($R^2_{adj,marg} = 0.83$). In addition to the functional trait and stoichiometric shifts, canopy and topsoil
$\delta^{15}N$ decreased on both continents, with a stronger decrease for canopy than for topsoil ($R^2_{adj,marg}$
respectively 0.92 and 0.43 for canopy and soil).

Discussion
The general characteristics, averaged for the altitudinal clusters, show that the vegetation structure
is different along both transects. The high variability in these structural characteristics as such, is
potentially caused by the relatively small plot size. However investigating the variation in
structural characteristics is beyond the scope of this research. Other research efforts, targeting
structural characteristics and carbon stocks in specific, use plot sizes of 1 hectare, as set forward
by the RAINFOR protocol, in tropical forests worldwide (Phillips et al., 2009). As such, the
differing carbon stocks probably do not integrate important stochastic events (e.g. tree fall) from
the forest along both slopes. However, interestingly enough the lower average carbon stocks and
higher number of trees in the upper two Rwandan clusters in comparison to the Ecuadorian forests.



This contrasts to what has been reported from large scale forest monitoring networks across the
lowland forests of Amazon and the Congo basin (Lewis et al., 2013). More research in larger plots,
including dynamics and productivity should validate if this is a consistent observation in highland
forest on both continents. On the other hand, the lower species number on the African transect fits
well within the recent findings of a pantropical study, reporting a lower tree species diversity in
the African tropical forest (Slik et al., 2015b). More importantly for this paper, the air temperature
decrease with elevation is highly similar on both transects, which means that we can validly assess
similar temperature-driven responses of both forest functional composition and the underlying
nutrient dynamics. The high collinearity in the trait datasets corresponds well to known trade-offs
described as the "leaf economics spectrum" (LES); basically a leaf-level trade-off between leaf
construction cost, i.e. low specific leaf area (SLA), leaf nitrogen content (LNC) and leaf
phosphorus content (LPC); and photosynthetic efficiency, i.e. high SLA, LNC and LPC (Wright
et al., 2004). LNC, LPC and SLA showed a highly significant decrease with altitude (Fig. 1 and
Table 2), indicating a functional shift towards more nutrient conservative species communities at
higher altitudes on both transects. Indeed, leaves at lower altitudes with high LNC, LPC and SLA
and hence a more efficient photosynthetic apparatus and rapid turnover, are replaced by leaves
with low LNC, LPC and SLA values at higher altitudes. Along the transects, both topsoil and
canopy leaves showed decreasing $\delta^{15}N$ values with increasing altitude (Fig. 1), inferring a more
closed N-cycle with lower N availability at the higher altitudes of both transects. We've added
previous published work in South America, with similar temperature gradients by Asner et al,
Kitayama and Aiba and Van de Weg et al. to our transects (12, 16, 17; Fig. S3) to assess the
consistency of our observed trends. We deliberately only added the limited amount of studies
where community-weighted means were reported along a 'single mountain range system', hence

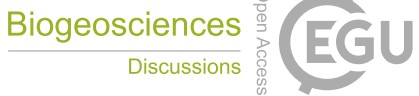



neglecting a recent and relevant contribution from Asner et al. (Asner and Martin, 2016). As argued
by Tanner et al. (Tanner et al., 1998), in studies of general altitude patterns from different mountain
ranges, elevational trends might be obscured by a range of local environmental characteristics,
such as soil type and orientation. Forests with low, intermediate and high nutrient concentrations
can be found globally at any altitude, hence neglecting the effect of local characteristics might lead
to wrong conclusions. This is also shown by the different intercepts, or the vertical *shift* in trends
in fig S3. Our comparison showed that the decreasing trend in LNC (mass basis) was consistent
with the other studies from South-America (Asner et al., 2016b; Van de Weg et al., 2009), but not
with South-East Asia where no significant trend was found (Van de Weg et al., 2009). However,
leaf mass area (LMA; the inverse of SLA) of all studies showed a similar, increasing trend with
elevation. LPC shows a strong and significant trend along both transects in this study, while the
other studies report no significant trend. This is consistent with the meta-analysis presented by
Tanner et al., which shows consistent negative LNC trends on 'same mountain' studies and
inconsistent LPC trends (Tanner et al., 1998). A recent effort on a larger scale in Peru has shown
that LES trade-off between LNC-LPC or SLA-LPC is indeed decoupled by climatic and
geophysical filters, while the leaf SLA-LNC trade-off is more robust (Asner et al., 2016a). Of the
included studies in Fig. S3, only Van de Weg et al. assessed N:P ratio, and although no significant
trend was found, they reported that N:P ratio was lowest in the highest sites (Van de Weg et al.,
2009). Additionally, decreasing N:P ratios have also been reported on other transects on the Andes
(Fisher et al., 2013; Soethe et al., 2008), and recently in Peru using airborne imaging spectroscopy
(Asner et al., 2016a).



In addition to our community-level traits (decreasing LNC, increasing C:N and N:P ratio), the
decreasing $\delta^{15}N$ values on both continents (fig. 1) are another strong indication of the limited
access of the upper forests to bio-available N. This data is only indicative and we cannot conclude
that there is actual limitation. However, the trends are interesting and seem to support the existing
paradigm that tropical forests shift from P to N limitation in transition from lowland to montane
tropical forest (Townsend et al., 2008). This is also reflected in the stoichiometric shifts, as the
C:N increase dominates over the C:P increase along the transects, leading to a general decrease of
N:P with increasing elevation (fig. 1). Hence plants build in relatively less N compared to P in
canopies at higher altitudes. The higher soil $\delta^{15}N$ values along the lower part of the Rwanda
transect suggests a more open N-cycle compared to the lower part of the Ecuadorian transect. This
corroborates with a recent finding of very high N losses at 1900 masl at the Rwanda site (Rütting
et al., 2014), and the finding of high retention of bio-available N in Chilean Andisols (Huygens et
al., 2008). Further research is needed to explain the notable divergence in soil and foliage $\delta^{15}N$
along the Ecuadorian transect. As previously reported this can be due to different degrees of
mycorrhizal infection (Hobbie et al., 2005), different mycorrhizal association types (Craine et al.,
2009) or species-specific preferences for different forms of nitrogen (Kahmen et al., 2008).

By characterizing both community functional traits and canopy and soil $\delta^{15}N$, the data of these
both transects is consistent with a decreasing availability of soil N as elevation increases, which is
here for the first time confirmed for the African continent.  We suggest the reduced N availability
to be caused by an indirect temperature effect on the N-cycle, consistent with observations from a
direct fertilization experiment (Fisher et al., 2013). Lower temperatures slow down
depolymerization and N mineralization processes, hence also N bio-availability, thereby invoking



changes in the functional plant communities along the transects (Coûteaux et al., 2002; Marrs et
al., 2013). Future global change will most likely distort this evolutional conditions for N
availability; both directly via increased reactive N deposition (Galloway et al., 2008; Hietz et al.,
2011) and indirectly via a temperature effect on N mineralization in tropical forests. This raises
questions on the future of plant species within the already threatened montane tropical forest
biome, where higher N availability and temperature increase might distort the existing ecological
niches and in turn also increase N-losses. Further research should therefore focus on process-based
knowledge of N and P cycle dynamics along such transects to further assess if the availability is
actually limiting the ecosystems. These observations also have repercussions for carbon fluxes:
since nutrient availability exerts a stronger control on NEP than on gross primary production (GPP)
(Fernández-Martínez et al., 2014), it is likely that the $CUE_e$ will be lower at higher altitudes. It has
been hypothesized that this decrease in $CUE_e$ is due to an increased investment of photosynthates
to non-biomass components, such as root symbionts for nutrient mining and root exudates, in
expense of net primary production (NPP) (Vicca et al., 2012). However, recent empirical evidence
has shown for one transect in the Andes, that a decrease in GPP with increasing altitude is not
accompanied by a trend in CUE (Malhi et al., 2016). More work on carbon budgets along
elevational transects is needed to fully understand the role of N and P availability and its interaction
with climate gradients for the tropical forest carbon cycle.

**Conclusions**
Altogether, this study evidences parallel functional shifts with a similar direction and magnitude
along two comparable elevation gradients, in tropical forests on two different continents. The data
suggests, in two different ways, that this shift is caused by temperature-driven response of nutrient

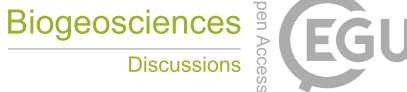

availability. This advocates the explicit need for implementing nutrient dynamics and availability
in ecosystem models. With the first data on an altitudinal transect in Central Africa, this work adds
to the existing set of elevational transects in the tropics, but more transects are needed, especially
in Africa, to validate a universal response of tropical forests. Furthermore, work on process-based
nutrient dynamics is important to unravel the importance of different global change factors for
both forest basins.

**Supplementary information**
**Fig. S1** Overview map
**Fig. S2** Structure and correlations of the trait data
**Fig. S3** Trends in community-level traits of previously reported studies
**Table S1** Coordinates, elevation and cluster membership of the different plots on both transects
**Table S2** Summary of the plot-level characteristics

**Author contributions**
M.B., H.V. and P.B. developed the project; M.B, M.D., S.B., C.T. and D.V. carried out the field
work and analyzed the data. All authors contributed to the ideas presented and edited the
manuscript.

**Competing interests**
The authors declare that they have no conflict of interest.



**Acknowledgments**

This research has been supported by the Belgian Development Cooperation through VLIR-UOS.

VLIR-UOS supports partnerships between universities and university colleges in Flanders

(Belgium) and the South looking for innovative responses to global and local challenges.

Visit www.vliruos.be for more information. We also thank BOS+ Tropen, Mindo Cloud Forest

Foundation and the Rwanda Development Board for the logistical support; Fidel Nyirimanzi and

Nicanor Mejía for their botanical expertise.



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



**Table 1** General characteristics, vegetation structure, climate (mean annual temperature (MAT) and mean annual precipitation (MAT)) and soil characteristics of the altitudinal clusters on both transects. Number of trees and species (in the 40 by 40 m plots), basal area (BA), mean tree height (MTH) and above-ground carbon (AGC) are averages per plot ± the standard deviation on the plot-level results, based on the inventories.

| | Cluster | Altitude (masl) | Number of trees per plot | Number of species per plot | BA ($m^2 ha^{-1}$) | MTH (m) | AGC (Ton C $ha^{-1}$) | MAT (°C) | MAP (mm) | Soil parent material | Soil classification |
|---|---|---|---|---|---|---|---|---|---|---|---|
| Ecuador | 1 | 406 ± 10 | 86 ± 13 | 30 ± 2 | 25 ± 3.1 | 18.3 ± 1 | 96 ± 19 | 23.7 | 3720 | Lahars | Andisol |
| | 2 | 1068 ± 25 | 84 ± 38 | 39 ± 15 | 33 ± 9.4 | 16.9 ± 0.7 | 140 ± 25 | 20.0 | 3227 | Lahars | Andisol |
| | 3 | 1871 ± 79 | 69 ± 11 | 31 ± 2 | 33 ± 11 | 13.5 ± 1.3 | 112 ± 57 | 17.5 | 1619 | Redbed volcaniclastics | Andisol |
| | 4 | 3217 ± 21 | 90 ± 25 | 18 ± 2 | 49 ± 11 | 13 ± 1.4 | 161 ± 34 | 10.9 | 1241 | Granitic/acid | Andisol |
| Rwanda | 1 | 1760 ± 66 | 70 ± 18 | 21 ± 4 | 34 ± 4 | 13.9 ± 0.7 | 121 ± 11 | 17.6 | 1518 | Shale and Quartzite | Inceptisol/Ultisol |
| | 2 | 2200 ± 64 | 71 ± 18 | 18 :± 3 | 45 ± 9 | 14.3 ± 0.3 | 179 ± 27 | 15.9 | 1628 | Shale and Quartzite | Inceptisol/Ultisol |
| | 3 | 2512 ± 37 | 122 ± 60 | 11 ± 1 | 31 ± 9 | 12 ± 1 | 99 ± 36 | 14.7 | 1716 | Shale and Quartzite | Inceptisol/Ultisol |
| | 4 | 2844 ± 77 | 109 ± 56 | 8 ± 2 | 34 ± 4 | 11.5 ± 0.5 | 89 ± 10 | 12.9 | 1835 | Shale and Quartzite | Inceptisol/Entisol |





**Table 2** Fixed effects estimates (altitude in km masl) for the different canopy-level response
variables; leaf nitrogen content (LNC), inverse C:N ratio, specific leaf area (SLA), leaf phosphorus
content (LPC), N:P ratio, canopy and topsoil δ15N, along with the estimated marginal and
conditional R2adj (sensu Nakagawa and Schielzeth (5)). The interaction term for altitude x transect
was not significant in any case, and was hence not retained in any model.

| Response | Effect | Estimate | Standard Error | Marginal $R^2_{adj}$ | Conditional $R^2_{adj}$ |
|---|---|---|---|---|---|
| LNC (%) | Ecuador Intercept | 3.04 | 0.11 | 0.83 | 0.85 |
| | Rwanda Intercept | 3.65 | 0.1 | | |
| | Altitude | -0.59 | 0.06 | | |
| N:C | Ecuador Intercept | 0.066 | 0.0026 | 0.87 | 0.91 |
| | Rwanda Intercept | 0.074 | 0.0023 | | |
| | Altitude | -0.013 | 0.001 | | |
| SLA (cm$^2$ g$^{-1}$) | Ecuador Intercept | 175.63 | 11.81 | 0.83 | 0.95 |
| | Rwanda Intercept | 173.06 | 10.41 | | |
| | Altitude | -36.47 | 5.91 | | |
| LPC (%) | Ecuador Intercept | 0.158 | 0.010 | 0.68 | 0.87 |
| | Rwanda Intercept | 0.172 | 0.008 | | |
| | Altitude | -0.025 | 0.005 | | |
| N:P | Ecuador Intercept | 20.67 | 0.87 | 0.60 | 0.71 |
| | Rwanda Intercept | 24.07 | 0.77 | | |
| | Altitude | -1.82 | 0.44 | | |
| δ$^{15}$N Canopy (‰) | Ecuador Intercept | 4.06 | 0.509 | 0.92 | 0.96 |
| | Rwanda Intercept | 8.25 | 0.45 | | |
| | Altitude | -2.63 | 0.26 | | |
| δ$^{15}$N Topsoil (‰) | Ecuador Intercept | 5.62 | 1.21 | 0.43 | 0.86 |
| | Rwanda Intercept | 8.11 | 1.07 | | |
| | Altitude | -1.34 | 0.61 | | |







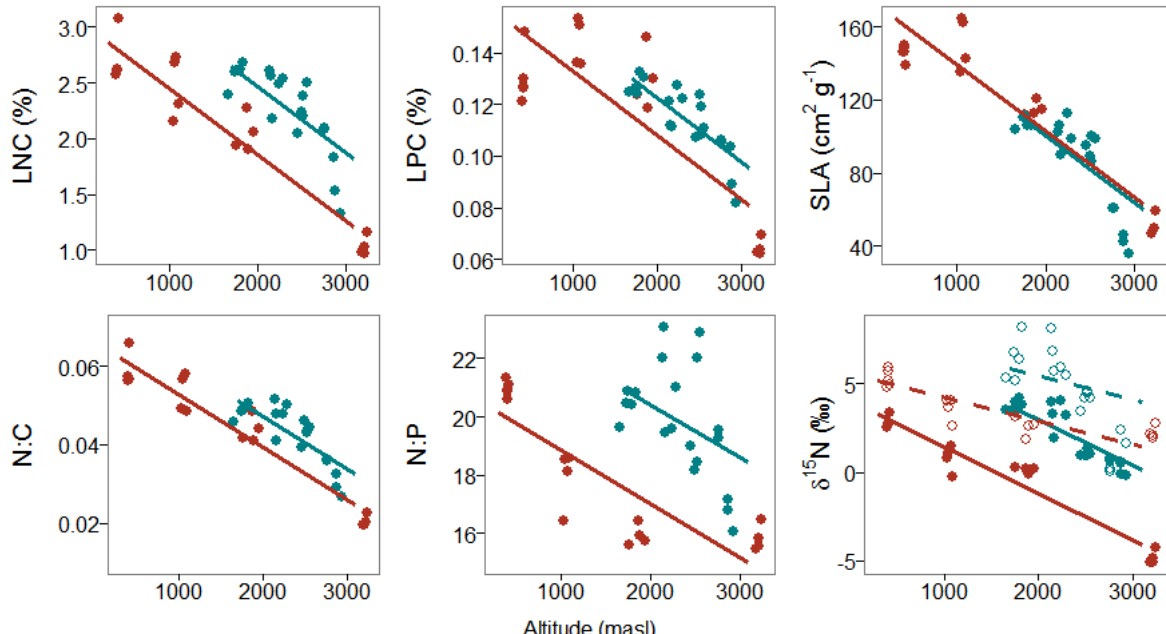

**Fig. 1. Trends in community-level functional traits and leaf (full line, closed circles) and topsoil (dashed line, open circles) $\delta^{15}$N of the elevation transects in Ecuador (red) and Rwanda (blue).** Leaf nitrogen content (LNC), leaf phosphorus content (LPC), specific leaf area (SLA), and leaf N:C, P:C and N:P ratio decrease with increasing altitude on both transects. Both transect showed decreasing values of $\delta^{15}$N, providing additional evidence for a more closed N-cycle with increasing altitude. Lines represent the fixed altitude effects in the respective statistical models for both Ecuador (red, 400-3200 masl) and Rwanda (blue, 1600-3000 masl).