# Peer review of "Parallel functional and stoichiometric trait shifts in South-American and African forest"

_Biogeosciences, 2017_

## Referee Comment (RC1) · J. Mayor (Referee) · 24 Apr 2017

General Comments:

I found the paper well written and of general importance to the scientific community given the rarity of elevational transects in tropical forests. My concerns with the current version are generally technical, based on a few somewhat easily modified statistical and linguistic approaches that would greatly improve the manuscript.

(1) Consider that altitude itself is not biologically meaningful, only factors that change with altitude (temperature, pressure, UV). So, in essence you are statistically comparing a proxy for climatic variables that influence plant and soil processes. This is cer-

tainly acceptable when climatic variables are not representative, as in the case where climate stations are too remote (or at the wrong elevation) from study plots. However, if you believe your MAT/MAP data reported in Table 1 are real then those metrics would be more sensible to use than altitude.This is because 2000 m elevation means nothing to a plant, and does not represent the same temperatures at different latitudes, aspects, distances from oceans, etc.

(2) Improving the statistical approach is unlikely to change the authors overall conclusions but currently does not represent the best practices in the field. I outline several specific areas for improvement below.

(3) The authors use functional characteristics and ecosystem function interchangeably. I believe this to be invalid but acknowledge that there are many whom would disagree with this semantic distinction. I agree that sometimes SLA or LNC is related to leaf function (e.g., photosynthetic rates) but it does not necessarily represent ecosystem function. Similarly, leaf and soil ðİŻ£15N values can integrate ecosystem N cycles but the authors to not demonstrate firm understanding of the caveats and alternatives. Specifics below.

Specific Comments:

L140 Consider that lme4 is the preferred package as nlme is no longer supported and therefore uses older estimators.

L141 Including the interaction and then dropping it from final models based on P values is in essence manual stepwise selection, which is generally not recommended when selecting from nested models. Consider using AIC or just leaving the interaction in. Also, there is no description of how these P values were estimated from an nlme model and this is a topic of great debate in the literature as determining the denominator degrees of freedom for random error variance is not a straightforward exercise. For instance, when using mixed effect models and P values the authors need to report something to the effect of: "hypothesis testing was evaluated using likelihood ratio

tests with type III ANOVA Satterthwaite approximation of degrees of freedom in the lmerTest package".

L142 The authors mention they assessed linearity (normality?) but do not discuss what other model fitting metrics they evaluated. Heteroscedascity? LPC certainly looks non-normally distributed, thus in violation of the pearson correlation statistics that are reported on Fig. S2.

Fig. S2 While on the subject, correlation among variables that contain the same variables will always be correlated. Therefore, reporting correlation statistics (with P values!) between LNC, C:N, and N:P is nonsensical.

L152 "Climatic conditions were similar" is simply not true. Rwanda gets wetter and the Ecuador gets drier with increasing elevation. This is a major problem with elevational transects in the tropics as sites that get wetter at high elevations also receive less solar inputs and will typically contain less foliar N as a result of reduced photosynthetic variability (not to mention either greater hydrological or gaseous outputs of soil N). It is true that both transects get colder but they do so within very different temperature ranges and magnitudes (12.8° vs. 4.7° changes that barely overlap) owing to the vastly (∼300%) different altitudinal ranges among transects (2811 vs. 1084 m). How can these authors claim they are similar? Where did the temperature data come from? What are the latitude and longitudes of the sites?

Instead, the author should refer to the similar adiabatic declines (e.g., 219 m per -1° in E, 230 m per -1° in R), then graphically depict temperatures of the plots; same goes for Fig. S3. This is one of the most important comments I provide on this manuscript.

L163 Consider explicitly examining the difference between foliar and soil ðİŻ£15N as this may better represent aspects of the N cycle (shameless plug: see Mayor et al. 2016, Eco. Lett. 18 for an in depth discussion of the patterns among tropical ðİŻ£15N values).

L181 The authors should refer to the similar adiabatic declines (e.g., 219 m per -1° in E, 230 m per -1° in R.

L203 One could also find elevation transects where the parent material changes from low to high along the same mountain as well. Also, you imply that your transects have consistent geology, aspect, slope etc. when they clearly do not based on Table 1 and Fig. S1. So the position you invoke, based on Ed Tanner's defense of low replication, is invalidated by your own data. In addition, there was a recent large scale global elevation gradient study published in Nature (another shameless plug) that suggests within-regional variability is negligible when comparing cross-continents. Given this, your selection of datasets based used in Fig. S3 to those with only single mountain transects may limit your conclusions and does not appear well justified.

L206-207 So Van de Weg data is from both SA and SE Asia? Why don't the lines reflect that on Fig. S3? Why don't the lines simply state the location of the transects rather than the authors?

L222 As mentioned, what may be more informative is the enrichment of plants relative to soil $\delta^{15}N$ values ($\triangle^{15}N_{plant-soil}$) is increasing at both sites. This may indicate either greater fractionating pathways during N uptake/translocation or a shift from one N form to another, rather than simply a more closed N cycle. Also, the Rwanda soil line does not appear to fit those data. Were any nonlinear lines compared?

Refs to consider here:

D. N. L. Menge, W. Troy Baisden, S. J. Richardson, D. A. Peltzer, M. M. Barbour, New Phytol , no- (2011). E. A. Davidson, C. J. R. de Carvalho, A. M. Figueira, F. Y. Ishida, et al., Nature 447, 995-8 (2007). J. Mayor, M. Bahram, T. Henkel, F. Buegger, et al., Ecol Lett 18, 96-107 (2015). C. Averill, A. Finzi, Ecology 92, 883-91 (2011). E. Bai, B. Z. Houlton, Global Biogeochem. Cycles 23, GB2011 (2009).

L234 You mean to say "different degrees of dependence upon ectomycorrhizal fungi"

in particular. (Although there is no discussion about the mycorrhizal type of your tree communities, I would assume they are AM which do not appreciably fractionate). Also, there is no talk about why higher soil 15N suggests and open N cycle when there is a large body of literature that clearly discusses that it may be due to greater gaseous N losses to denitrifiers.

L240 "Confirmed"? No, not confirmed, as you say above, only suggestive of declining bioavailable N.

L242 Here is a statement that would be much stronger if you were actually looking at the same temperature ranges across all transects in Fig. S3.

L245 What is "this evolutionary conditions for N" mean?

Technical Corrections:

L53 Extra comma.

L156 "ajd" subscript incorrect.

L160 Spacing between R2 is off.

L227 What does "C:N increase dominates over the C:P increase along the transects"
mean? Also, Fig. 1 lists N:C, not C:N, and there is no C:P. . .

L228 Consider replacing "build in" with "incorporate" or "use".

L239 Delete "both".

L508 Fix 15N superscript.

L509 Fix R2adj subscript.

L517 It would be useful to those conducting meta-analyses if these raw data were made available.

---

## Referee Comment (RC2) · Anonymous Referee #2 · 28 Jul 2017

General comments

The manuscript by Bauters et al. presents an interesting analysis based on a detailed survey of tree species occurrences and their functional traits along two altitudinal transects. The main findings concerning the similar shifts in community weighted means of the key leaf traits for American and African altitudinal transects is stunning and of potential interest to a broad audience. Having that said, there are several issues that need to be improved.

Specific comments

My major point concerns a decoupling between results and theoretical background.

[Figure]

The reasoning behind the study needs to be better developed and introduced.

Understanding the links between nutrient availability and species composition or biomass production is certainly a key for tropical forests. Nevertheless, it is not fully clear how particularly this study contributes to this understanding. For instance, the first two lines in Abstract state: "Elevational gradients are an empirical tool to assess long-term forest responses to environmental change. We studied whether functional composition of tropical forest along elevational gradients in South America and in Africa showed similar shifts." I do not understand how these sentences are connected. If the shifts are similar, what can this result tell us about forest responses to environmental change?

Similarly, lines 33-34: "This highlights the importance of nutrient availability for tropical forest in a changing world."

I feel that this sentence is not a sufficient explanation for the main result found here. Why is nutrient limitation important and how this study contributed to this conclusion?

Moreover, as nitrogen concentration changes with both elevation and altitude, how can we know that nutrient concentration (and not temperature) is the key driver of species composition?

I recommend authors to better develop and explain the hypotheses tested.

Second, it is being argued that phosphorus rather than nitrogen is the key nutrient limiting biomass production (and perhaps also species distribution) in tropical forests (Aragao et al. 2009, Quesada et al. 2009). Why this study focuses on nitrogen concentration only? This issue should be discussed.

Aragão, L. et al. 2009. Above-and below-ground net primary productivity across ten Amazonian forests on contrasting soils. – Biogeosciences 6: 2759–2778. Quesada, C. A. et al. 2009. Regional and large-scale patterns in Amazon forest structure and function are mediated by variations in soil physical and chemical properties. – Biogeosci.

Discuss. 6: 3993–4057.

Third, If "Elevational gradients are an empirical tool to assess long-term forest responses to environmental change" (first line of Abstract), how general these results can be? E.g. could we find similar pattern along latitudinal transects?

Lastly, the first paragraph of Discussion is difficult to read. I recommend starting with the main results, how are these results linked with the predictions and what is the possible implication. It is unclear what "the general characteristics" means and what "vegetation structure" means and what "structural characteristics" means

Technical comments:

Line 53: extra ","

Line 95: Unclear whether tree height or carbon stock was estimated using pantropical relationship. Perhaps delete the comma after "(AGC)"?

Line 509: "2" in upper case (R2adj)

―――――――――――――――――――――

---

## Author Comment (AC1) · 7 Aug 2017

**Response to RC1**

General Comments: I found the paper well written and of general importance to the scientific community given the rarity of elevational transects in tropical forests. My concerns with the current version are generally technical, based on a few somewhat easily modified statistical and linguistic approaches that would greatly improve the manuscript.

**We want to thank the reviewer for his appreciation, and his thorough, constructive and qualitative review. We have addressed the comments in bold in this response, and have adapted the MS as noted below. All the edits are indicated in the manuscript (MS) text with track changes.**

(1) Consider that altitude itself is not biologically meaningful, only factors that change with altitude (temperature, pressure, UV). So, in essence you are statistically comparing a proxy for climatic variables that influence plant and soil processes. This is certainly acceptable when climatic variables are not representative, as in the case where climate stations are too remote (or at the wrong elevation) from study plots. However, if you believe your MAT/MAP data reported in Table 1 are real then those metrics would be more sensible to use than altitude. This is because 2000 m elevation means nothing to a plant, and does not represent the same temperatures at different latitudes, aspects, distances from oceans, etc.

**We fully agree with the referee on this point, including direct variables instead of the elevation proxy is highly interesting for elevational transects. There are 2 reasons why we did not do this in the manuscript, and by which we would also defend the current version of the manuscript:**

1. **Most important: as the referee rightfully states, there are variables other than precipitation or temperature that covary with elevation (such as irradience, cloud cover, UV...). We merely use elevation in the MS as a proxy because we feel that using all these covarying independent variables would at this point be too suggestive, given the limited size of the dataset (two transects) and the limited orthogonality of the setup. However, we do report MAT and MAP (and the coordinates of the sites in Table S1) to allow future meta-analysis with more transects/plots to disentangle these effects, and as a background for the study.**
2. **The data reported in the tables is WorldClim data (we have also more clearly stated that in the new version of the MS), and indeed the climate stations are both remote and at different locations. One of the notorious challenges in higher elevation is to assess fog deposition, and which probably contributes a lot to the hydrological cycle of high elevation forest. We believe that for before including MAP and MAT in the analyses, advances in this field and in the quantification of this phenomenon are needed, since it is likely that MAP is heavily underestimated at higher elevations.**

(2) Improving the statistical approach is unlikely to change the authors overall conclusions but currently does not represent the best practices in the field. I outline several specific areas for improvement below.

**See below for specific answers. We are grateful for these comments and have implemented them in the new MS version, although indeed the overall conclusions did not change, as the referee rightfully states.**

(3) The authors use functional characteristics and ecosystem function interchangeably. I believe this to be invalid but acknowledge that there are many whom would disagree with this semantic distinction. I agree that sometimes SLA or LNC is related to leaf function (e.g., photosynthetic rates) but it does not necessarily represent ecosystem function. Similarly, leaf and soil d15N values can

integrate ecosystem N cycles but the authors to not demonstrate firm understanding of the caveats and alternatives. Specifics below.

**This is a thoughtful comment, which we have already adapted in the new MS. The MS should be objective and correct in the first place, and we would like to avoid any confusion in the text. We have now more explicitly added a section on functional traits in the introduction, which states more clearly that these are 'only' proxies. See also below for more detailed answers.**

Specific Comments:

L140 Consider that lme4 is the preferred package as nlme is no longer supported and therefore uses older estimators.

**Ok, thank you for this comment. We have now reprocessed the data with the lme4 package.**

L141 Including the interaction and then dropping it from final models based on P values is in essence manual stepwise selection, which is generally not recommended when selecting from nested models. Consider using AIC or just leaving the interaction in. Also, there is no description of how these P values were estimated from an nlme model and this is a topic of great debate in the literature as determining the denominator degrees of freedom for random error variance is not a straightforward exercise. For instance, when using mixed effect models and P values the authors need to report something to the effect of: "hypothesis testing was evaluated using likelihood ratio tests with type III ANOVA Satterthwaite approximation of degrees of freedom in the lmerTest package".

**This was not well described, the referee is right. In practice we used AIC, BIC and the default ML ratio testing in the anova.nlme function, which is not documented very well in the package. I see now that lme4 has better documentation on this, hence we have shifted to that and better described the methods used in a next MS version. As stated above, we have now re-analysed the data with the lme4 package, and better discribed the methodology (significant fixed effect) using the lmerTest package. We now also included an extra table in supplementary information with the models including the interaction term, along with their respectives AIC scores.**

L142 The authors mention they assessed linearity (normality?) but do not discuss what other model fitting metrics they evaluated. Heteroscedascity? LPC certainly looks non-normally distributed, thus in violation of the pearson correlation statistics that are reported on Fig. S2.

**Yes, normality and heteroscedascy of the residuals, qualitatively via qqplots and residual plots. We now more clearly state this in the MS. Please see answer on comment below for the correlation matrix.**

Fig. S2 While on the subject, correlation among variables that contain the same variables will always be correlated. Therefore, reporting correlation statistics (with P values!) between LNC, C:N, and N:P is nonsensical.

**The referee is right. Actually, the main information we wanted to give the reader was the structure of the trait data sets. And we felt that reporting the correlation statistics in supplementary information, gives the reader the opportunity to gain insight. Mainly when considering the link to ecosystem functionality, the links between traits representing the so called 'trade-offs' are in our opinion valuable to report. In the new version, we have log-transformed the data prior to reporting the correlation statistics. This did still not render normal distributions for some of the parameters, so we now report Spearman's correlation tests, which do not rely on the normality assumption.**

L152 "Climatic conditions were similar" is simply not true. Rwanda gets wetter and the Ecuador gets drier with increasing elevation. This is a major problem with elevational transects in the tropics as sites that get wetter at high elevations also receive less solar inputs and will typically contain less foliar N as a result of reduced photosynthetic variability (not to mention either greater hydrological or gaseous outputs of soil N). It is true that both transects get colder but they do so within very different temperature ranges and magnitudes (12.8◦ vs. 4.7◦ changes that barely overlap) owing to the vastly (∼300%) different altitudinal ranges among transects (2811 vs. 1084 m). How can these authors claim they are similar? Where did the temperature data come from? What are the latitude and longitudes of the sites? Instead, the author should refer to the similar adiabatic declines (e.g., 219 m per -1◦ in E, 230 m per -1◦ in R), then graphically depict temperatures of the plots; same goes for Fig. S3. This is one of the most important comments I provide on this manuscript.

**The latitude and longitudes are listed in Table S1 of the original MS. We agree, and acknowledge in the manuscript, that the transects are different in range for elevation and temperature (see e.g. L 89-90), there was unfortunately no other option in the field (neither in any part of eastern DRC or eastern central Africa for that matter!). Additionally, line 152 completes to '*Climatic conditions were similar, with a highly consistent temperature gradient (Fig S3, Table 1), and similar mean annual precipitation in the concurring altitudinal ranges*.' We do acknowledge that the formulation is a bit misfortunate, but we actually mainly wanted to focus on the very similar adiabatic lapse rate '*highly consistent temperature gradient*' (as we refer to the parallels in fig S3), as noticed by the referee indeed. Additionally, the Ecuadorian transect is 'enveloping' the Rwanda transect, and the amounts of rainfall and temperature conditions within the overlapping altitudinal zone are similar. This boils down to the discussion above; we fully agree that altitude is only a proxy, but we do believe that directly including more climatic predictors instead would be more useful for a larger meta-analyses. But exactly for that reason, empirical research results on 'smaller' datasets (which are very labour and time intensive to acquire, and therefor rare in the tropics) have to get published.**

L163 Consider explicitly examining the difference between foliar and soil d15N as ˙ this may better represent aspects of the N cycle (shameless plug: see Mayor et al. 2016, Eco. Lett. 18 for an in depth discussion of the patterns among tropical d15N values).

**Please see answer below (on comment of L 234) for an elaborate response on this.**

L181 The authors should refer to the similar adiabatic declines (e.g., 219 m per -1◦ in E, 230 m per -1◦ in R.

**We now do this explicitly in the results section.**

L203 One could also find elevation transects where the parent material changes from low to high along the same mountain as well. Also, you imply that your transects have consistent geology, aspect, slope etc. when they clearly do not based on Table 1 and Fig. S1. So the position you invoke, based on Ed Tanner's defense of low replication, is invalidated by your own data. In addition, there was a recent large scale global elevation gradient study published in Nature (another shameless plug) that suggests within-regional variability is negligible when comparing cross-continents. Given this, your selection of datasets based used in Fig. S3 to those with only single mountain transects may limit your conclusions and does not appear well justified.

**Good point. We have reformulated this in a new version. However, the point raised is debatable in our opinion, and we would like to keep the meta-analyses as is in the new version. 1) There are other environmental variables than parent material that are also important when not working with single mountain studies (seasonality, atmospheric deposition, continentality). Additionally, although the parent material is 'varying' in Ecuador along the slope, the soils are all Andisols and are from volcanic**

**origin. In our opinion, the best proof that we should restrict in this study to single mountain studies, is the shift in intercept between both of our own two transects (although the slopes are parallel). 2) Our own transects are 'same mountain' transects, the MS might lose focus with a more extensive meta-analyses using 'non-same mountain' studies (and this has been reported already by Asner et al. 2016 GBC), and 3) much as can be seen in the latter paper, there is simply no data (not from same mountain studies, but also not in a broader approach) on African forests. So would a more elaborate meta-analysis genuinely contribute much more to our story, given the strong focus we have on the African continent currently? (We don't think so)**

L206-207 So Van de Weg data is from both SA and SE Asia? Why don't the lines reflect that on Fig. S3? Why don't the lines simply state the location of the transects rather than the authors?

**Thanks for this, this is wrong in the MS. SE Asia should be Kitayama! We've added the locations in the legend now.**

L222 As mentioned, what may be more informative is the enrichment of plants relative to soil d15N values ( ˙ Δ15Nplant-soil) is increasing at both sites. This may indicate either greater fractionating pathways during N uptake/translocation or a shift from one N form to another, rather than simply a more closed N cycle. Also, the Rwanda soil line does not appear to fit those data. Were any nonlinear lines compared? Refs to consider here: D. N. L. Menge, W. Troy Baisden, S. J. Richardson, D. A. Peltzer, M. M. Barbour, New Phytol , no- (2011). E. A. Davidson, C. J. R. de Carvalho, A. M. Figueira, F. Y. Ishida, et al., Nature 447, 995-8 (2007). J. Mayor, M. Bahram, T. Henkel, F. Buegger, et al., Ecol Lett 18, 96-107 (2015). C. Averill, A. Finzi, Ecology 92, 883-91 (2011). E. Bai, B. Z. Houlton, Global Biogeochem. Cycles 23, GB2011 (2009).

**This was partly discussed in the previous MS version L232-236. See a more elaborate answer below.**

L234 You mean to say "different degrees of dependence upon ectomycorrhizal fungi" in particular. (Although there is no discussion about the mycorrhizal type of your tree communities, I would assume they are AM which do not appreciably fractionate). Also, there is no talk about why higher soil 15N suggests and open N cycle when there is a large body of literature that clearly discusses that it may be due to greater gaseous N losses to denitrifiers.

**We have responded here to all the concern raised by the referee on the d15N trends. The reviewer's comments on this are very valuable for this MS. We did include a more elaborate discussion on this in an earlier draft, but felt that it might render the discussion to suggestive. Given the parallel comments of the other referee on this, we have now more elaborately discussed the d15N patterns in a following MS version. First of all the soil patterns as such, with a more in-depth discussion on why decreasing d15N values suggest a closing N cycle. Additionally, the Δ15N (canopy-soil) patterns, with the work of especially Mayor et al. 2015 and Averill et al. 2011. We have now also changed the statistics for the d15N values. We considered each transect as such, and then fitted a mixed effect model to the d15N data, including the compartment it was measured in (soil or canopy) as a fixed effect, with interaction term with altitude. This allows us to elegantly discuss the divergence or convergence of soil-canopy d15N values (so, Δ15N) along both transects, given the significance of the interaction terms in both cases.**

L240 "Confirmed"? No, not confirmed, as you say above, only suggestive of declining bioavailable N.

**Ok, was indeed overstating! The sentence did not contribute a lot, so we deleted it.**

L242 Here is a statement that would be much stronger if you were actually looking at the same temperature ranges across all transects in Fig. S3.

**Please see answers above.**

L245 What is "this evolutionary conditions for N" mean?

**We have rephrased this. We meant that temperature ranges are likely to shift in the future, which will affect N cycling along slopes.**

Technical Corrections:

**Thanks for these!! We have adapted this in the new version**

L53 Extra comma.

L156 "ajd" subscript incorrect.

L160 Spacing between R2 is off.

L227 What does "C:N increase dominates over the C:P increase along the transects" ǎAˇˊlmean? Also, Fig. 1 lists N:C, not C:N, and there is no C:P. . .

L228 Consider replacing "build in" with "incorporate" or "use".

L239 Delete "both".

L508 Fix 15N superscript.

L509 Fix R2adj subscript.

L517 It would be useful to those conducting meta-analyses if these raw data were made available.

**Raw data was and is available in the supplementary information of the original submission.**

**Response on Revierwer 2**

General comments

The manuscript by Bauters et al. presents an interesting analysis based on a detailed survey of tree species occurrences and their functional traits along two altitudinal transects. The main findings concerning the similar shifts in community weighted means of the key leaf traits for American and African altitudinal transects is stunning and of potential interest to a broad audience. Having that said, there are several issues that need to be improved.

**We would like to thank referee 2 for the thoughtful and qualitative review. We have addressed the comments in bold in this response, and have adapted the MS as noted below. All the edits are indicated in the manuscript (MS) text with track changes.**

Specific comments

My major point concerns a decoupling between results and theoretical background.

The reasoning behind the study needs to be better developed and introduced.

Understanding the links between nutrient availability and species composition or biomass production is certainly a key for tropical forests. Nevertheless, it is not fully clear how particularly this study contributes to this understanding. For instance, the first two lines in Abstract state: "Elevational gradients are an empirical tool to assess long-term forest responses to environmental change. We studied whether functional composition of tropical forest along elevational gradients in South America

and in Africa showed similar shifts." I do not understand how these sentences are connected. If the shifts are similar, what can this result tell us about forest responses to environmental change?

**We have reformulated and rewritten the framework of the study in the new MS version, so that this is clearer. See also comment on point 3 raised by the referee. We use the elevational transects as space for time tools, where the limited spatial range subjects the forest to strong shifts in environmental – mainly climatic – conditions. We believe the main contribution of this MS is the report of highly parallel shifts across continents, of ecosystem properties which have been previously linked to ecosystem functioning. The reported shifts show that forest communities and biogeochemical cycles on both continents respond to the drastic change in some of the environmental variables along the slope (temperature). The observation of shifts in both biogeochemical proxies (d15N) and ecological functioning proxies (leaf traits) shows that the biogeochemical cycles are strongly linked to the ecological shifts.**

Similarly, lines 33-34: "This highlights the importance of nutrient availability for tropical forest in a changing world."

I feel that this sentence is not a sufficient explanation for the main result found here. Why is nutrient limitation important and how this study contributed to this conclusion?

**As written above, the shift in forest composition along the slope, with the reported N cycle proxies, shows that forest composition responds to the shifts in nutrient cycling along the transect.**

Moreover, as nitrogen concentration changes with both elevation and altitude, how can we know that nutrient concentration (and not temperature) is the key driver of species composition?

**We are a bit confused by this comment. The baseline of the story is that temperature shifts with altitude, which in turn affects the N cycling along the slope, and subsequently the species composition. We have now more elaborately described the state-of-the-art and the background on the used proxies in the new MS version, and hope that this is clearer.**

I recommend authors to better develop and explain the hypotheses tested.

Second, it is being argued that phosphorus rather than nitrogen is the key nutrient limiting biomass production (and perhaps also species distribution) in tropical forests (Aragao et al. 2009, Quesada et al. 2009). Why this study focuses on nitrogen concentration only? This issue should be discussed.

Aragão, L. et al. 2009. Above-and below-ground net primary productivity across ten Amazonian forests on contrasting soils. – Biogeosciences 6: 2759–2778. Quesada, C. A. et al. 2009. Regional and large-scale patterns in Amazon forest structure and function are mediated by variations in soil physical and chemical properties. – Biogeosci. Discuss. 6: 3993–4057.

**We have discussed this more in depth in the new MS version. We deliberately caution the use of 'limitation' in the MS, since there are different forms of nutrient limitation, and none are easily assessed in the field (see also reference). P is indeed considered the limiting nutrient for the bulk of the tropical forest. However, several reports show that forests shift from P to N limitation along elevational transects, for two reasons; 1) replacing the lowland oxisols, which are strongly weathered and P-depleted, with P-richer and geologically younger soils, and 2) slowing down of the N-cycle with lower temperatures in higher forest. Given the fact that we are indeed working on higher elevation forest, the nitrogen cycle is the main focus of the paper. See also discussion at L 224-228.**

1.      P. M. Vitousek, S. Porder, B. Z. Houlton, O. a Chadwick, Terrestrial phosphorus limitation : mechanisms, implications , and nitrogen – phosphorus interactions. *Ecol. Appl.* **20, 5–15 (2010).**

Third, If "Elevational gradients are an empirical tool to assess long-term forest responses to environmental change" (first line of Abstract), how general these results can be? E.g. could we find similar pattern along latitudinal transects?

**Elevational transects are an acknowledged tool for long-term forest responses, so called space-for-time experiments, and are in general better setups than latitudinal transects. Part of the reason is that the seasonality shifts with latitude (and the length of the growing season) along with other factors, which render latitudinal setups non-orthogonal. This is elaborately discussed in following introduction paper in a special issue on elevational transects in 2010:**

**Malhi, Y. *et al.* Introduction: Elevation gradients in the tropics: Laboratories for ecosystem ecology and global change research. *Glob. Chang. Biol.* 16, 3171–3175 (2010).**

Lastly, the first paragraph of Discussion is difficult to read. I recommend starting with the main results, how are these results linked with the predictions and what is the possible implication. It is unclear what "the general characteristics" means and what "vegetation structure" means and what "structural characteristics" means

**We agree with the referee and have reformulated in the new MS version.**

Technical comments:

**Thanks for these, we have adapted this in the new MS version.**

Line 53: extra ","

Line 95: Unclear whether tree height or carbon stock was estimated using pantropical relationship. Perhaps delete the comma after "(AGC)"?

Line 509: "2" in upper case (R2adj)

---

## Author Response (AR2)

**DEPARTMENT OF APPLIED ANALYTICAL AND PHYSICAL CHEMISTRY**
ISOTOPE BIOSCIENCE LABORATORY

Marijn Bauters

E   Marijn.Bauters@UGent.be
T   +32 9 264 60 06

Faculty of Bioscience Engineering
Coupure Links 653
B-9000 Gent
Belgium www.ugent.be

Biogeosciences editorial office
Copernicus publications

DATE
October 2017

page

Dear *Biogeosciences* editor,

We hereby submit our revised manuscript entitled "Parallel functional and stoichiometric trait shifts in South-American and African forest communities with elevation" to be considered for publication as a research article in *Biogeosciences*.

We would like to thank both reviewers and the editor for the thoughtful comments, fast handling, and positive and constructive assessments of our manuscript. We believe we have addressed and clarified each comment made by both reviewers. The detailed responses are below, and the line numbers refer to the revised manuscript with track changes, which is also below.

We feel that the revised manuscript has improved again and hope that its changes satisfy both reviewers and the editor.

Yours sincerely,

The authors

Responses to Reviewers

Dear Dr. Bauters and co-authors,
the two reviewers are satisfied with your revisions and have some suggestions for further improvement.

*We would like to thank the handling editor for the efforts made on this manuscript. The review process has been fast and constructive, and we feel that we were able to improve the manuscript substantially from the earlier version.*

In addition to their comments I ask you to take care of the following:
* Key words: I agree that „elevational gradient" is more appropriate than „altitudinal gradient", and suggest to delete „elevational transects", which overlaps significantly
*We have now adapted this in the new MS version (p1 L16).*

* l. 75 To my understanding elevational gradients typically do imply changes in growing season length. Please correct / delete this part of the statement.
*We have now adapted this in the new MS version (p3 L48).*

* l. 100 add reference
*We have now adapted this in the new MS version (p4 L74-75).*

* l. 242 your reasoning is hard to follow: why should it be nonsensical to report on and test for these abiotic drivers in your study?
*This was indeed poorly formulated. We have now adapted this in the MS. It would be sensical – interesting even- to test for abiotic drivers, but in our opinion it would be an overkill of the dataset that we report on. A more elaborate analysis with direct abiotic changes instead of altitude as explanatories would be interesting in a larger meta-analysis with a number of transects. Additionally, we don't have field estimates of these drivers in the sites (p10 L214-217).*

* Table 2: add unit for SLA
*We have now adapted this in the new MS version (p37 Table2).*

Please address these remaining issues and carefully correct the typos in a revision, and explain in your response letter how you have dealt with the comments and what changes you have made to the manuscript.

Best regards,
Michael Bahn

**Reviewer report 1**

In my opinion the authors have substantially improved the manuscript and addressed most of the concerns I had in the original version.
*We would like to thank the reviewer once more for all the efforts on this manuscript and apologize for the unclear points that remained after the previous review round. We feel that his/her time and efforts have substantially improved this MS.*

There are some issues with their response, however. Namely, the authors do not provide accurate line numbers for the new MS so that I can go check what they claim to have addressed. Do they give the old line numbers again or did they not go back and correct the line numbers during revision?
*The line numbers that we provided corroborated with the changes of the version with track changes that was provided with the response letter. We apologize for this unclarity and*

[Figure]

*acknowledge that we should have stated this explicitly in the response letter. The line numbers in the current revisions again refer to the revised MS with track changes below.*

Response to L203: What do you mean strong focus on Africa? Half of your data is from Ecuador and the datasets you compare to are from all over. My original comment was about your description of the methods used to select the single mountain transects that you compare your data to, which I think is poorly justified/explained in the MS. Saying it would be too much work to bring in more data is not a valid excuse. You once again cite an argument in Tanner 1998 that doesn't really support the decision you've made here - to use only single mountain transects because you can find high or low nutrient soils on any mountain? So why not use data from multiple mountains that would alleviate the chance that one was from an unusually low or high nutrient soil?

*We appreciate the reviewer's vision on this, and acknowledge that our formulated response may have been poorly expressed. We have now deleted Tanner's argument from the MS (p11 L234-237). But in the end, this might also be a point we will have to respectfully agree to disagree upon. As we explain in the MS, we add these studies where community-weighted averages of traits are reported from similar setups to compare to our study. We therefore only leave out the study by Asner et al., because he assembled community-weighted averages from plots all over the tropics and then looked at elevation as a driving factor for the changes noted in his dataset. As we acknowledge in the MS, this is an interesting study, but it uses a different approach as the one we, and others, use. If we were to include this study, where would we draw the line? Then we might include all the published data on community weighted average of any plot in the tropics. We prefer not to do this, not because of the workload, but because:*

1) *The goal of this study is to report new findings on two different elevation transects. To frame these findings, we compare them to existing reports with a similar setup. Doing a bigger meta-analysis is a different study, and would get the focus off our results.*
2) *As we stated in the previous response, there are other factors that might be confounding in such meta-analyses (atmospheric N deposition seasonality, etc.). Additionally, data of our two transects show the same slope but a different intercept. If we would pick plots randomly from both transects to add in such a meta-analysis; it would not be likely to find the same slopes (or maybe not even trends) while when looking at 'same mountain' trends the opposite is true.*

You again invoke the type or degree of mycorrhization to possibly explain your 15N trends without actually discussing if any small proportion of your tree species associate with EcM.
*We simply don't have data on EcM association, so we cannot confirm that this is/is not causing divergence in soil-canopy d15N. However, the shift in enrichment factor, if real, is likely caused by either a shift in plant N source, or a shift in mycorrhizal association. Hence we did not want to leave this "untouched" in the MS. We do acknowledge that we don't have EcM data in the MS.*

The manuscript will benefit greatly from careful editorial work as their are typically 3-4 typos per page.
*We apologize for this, and have now carefully re-edited the MS.*

**Reviewer report 2**

The authors have adequately addressed my previous comments, and I only have few minor comments.
*We would like to thank the reviewer for both review efforts. We acknowledge that his time and efforts have substantially improved the manuscript.*

[Figure]

[Figure]

The fact that the study is based on two transects only should be mentioned in abstract, so far it is unclear how many transects were used.
*We have now adapted this in the new MS version (p2 L26).*

Line 94 – "therefore" instead of "therefor"
*We have now adapted this in the new MS version (p4 L67).*

Line 280-281 – split into two sentences
*We have now adapted this in the new MS version (p12 L 248).*

Table 2 in Appendix should be perhaps Table S3?
*Thank you for this. We have now adapted this in the new MS version.(See supplementary information, and MS p15 L 323-324)*

The discussion is still a bit hard to follow for me and it would be helpful to stress the main findings and broader impact in the first paragraph.
*We have now included a short 'introductory' paragraph at the beginning of the discussion. (p9 L191-197)*

[revised manuscript text omitted]